# The Anti-Metastatic Properties of Glutathione-Stabilized Gold Nanoparticles—A Preliminary Study on Canine Osteosarcoma Cell Lines

**DOI:** 10.3390/ijms26136102

**Published:** 2025-06-25

**Authors:** Sylwia S. Wilk, Klaudia I. Kukier, Arkadiusz M. Michałowski, Marek Wojnicki, Bartosz Smereczyński, Michał Wójcik, Katarzyna A. Zabielska-Koczywąs

**Affiliations:** 1Laboratory of Experimental and Clinical NanoOncology, Department of Small Animal Diseases and Clinic, Institute of Veterinary Medicine, Warsaw University of Life Sciences, Nowoursynowska 159c, 02-776 Warsaw, Poland; sylwia_wilk@sggw.edu.pl (S.S.W.); klaudia_kukier@sggw.edu.pl (K.I.K.); arkadiusz_michalowski@sggw.edu.pl (A.M.M.); 2Faculty of Non-Ferrous Metals, AGH University of Krakow, Al. A. Mickiewicza 30, 30-059 Kraków, Poland; marekw@agh.edu.pl; 3Faculty of Chemistry, University of Warsaw, Pasteura 1, 02-093 Warsaw, Poland; b.smereczyns@student.uw.edu.pl (B.S.); mwojcik@chem.uw.edu.pl (M.W.)

**Keywords:** alpha 2 macroglobulin, canine osteosarcoma, gold nanoparticles, metastasis, migration, Simple Western

## Abstract

Osteosarcoma (OSA) is the most common primary bone malignancy in dogs, characterized by aggressive growth and high metastatic potential. Despite advances in treatment, the prognosis for affected animals remains poor, mainly due to metastatic disease. Metastasis is a complex process that involves forming new blood vessels in the primary tumor (angiogenesis), intravasation, the transport of cancer cells to other locations, extravasation, and the growth of cancer cells in the secondary site. Gold nanoparticles (AuNPs), due to their unique physicochemical properties, are considered promising tools in cancer therapy, both as drug delivery systems and potential anti-metastatic agents. Previously, it has been demonstrated that 500 µg/mL glutathione-stabilized gold nanoparticles (Au-GSH NPs) inhibit cancer cell extravasation—one of the steps of the metastatic cascade. This study aimed to evaluate the anti-metastatic properties of Au-GSH NPs through their influence on OSA cell migration, proliferation, and colony formation in vitro, as well as their antiangiogenic properties on the chick embryo chorioallantoic (CAM) model. Additionally, we investigated whether these effects are associated with changes in alpha-2-macroglobulin (A2M) expression, as it was previously demonstrated to play an essential role in the metastatic cascade. Au-GSH NPs significantly inhibited migration and colony formation in canine osteosarcoma cells (from OSCA-8, OSCA-32, and D-17 cell lines) at 200 µg/mL concentrations. Interestingly, at 500 µg/mL, Au-GSH NPs inhibited angiogenesis on the CAM model and cancer cell migration, but fewer colonies were formed. These results may be directly related to the higher efficiency of Au-GSH NPs uptake by OSA cells at the dose of 200 μg/mL than at the dose of 500 μg/mL, as demonstrated using Microwave Plasma Atomic Emission Spectroscopy (MP-AES). Moreover, this is the first study that demonstrates a significant increase in A2M expression in cancer cells after Au-GSH NPs treatment. This study provides new insight into the potential use of Au-GSH NPs as anti-metastatic agents in canine osteosarcoma, indicating that their anti-metastatic properties may be related to A2M. However, further in vitro and in vivo studies are needed to explore the molecular mechanism underlying these effects and to evaluate the clinical relevance of AuNPs in veterinary oncology.

## 1. Introduction

Osteosarcoma (OSA) is the most common bone malignancy in dogs, accounting for 80–90% of primary bone tumors [1]. It is characterized by aggressive growth and a tendency to metastasize, most often to the lungs, the bones, and other visceral organs such as the spleen and liver [2]. Metastasis is a complex process, which includes five key stages: 1. the formation of new blood vessels within the primary tumor, 2. local invasion and subsequent entry into the blood vessels (intravasation), 3. transport and lodging in the vessels at the secondary site, 4. exit from the vessels (extravasation) and migration, and 5. growth at the secondary site [3]. OSA treatment typically involves a combination of surgery, chemotherapy, and radiotherapy, although the prognosis remains poor [4]. Despite advances in diagnostics and therapy, the median survival time of dogs with osteosarcoma treated by amputation and following chemotherapy is approximately 284 days [5].

Nanomedicine shows great potential in oncology. Recent advances in nanobiotechnology have produced nanocarrier systems for targeted drug delivery in cancer therapy, enhancing treatment outcomes and reducing side effects [6]. Nanoparticle-based therapies act through either passive or active targeting. Passive targeting relies on the enhanced permeability and retention (EPR) effect, which occurs due to the abnormal structure of tumor blood vessels and impaired lymphatic drainage. These factors allow nanoparticles to accumulate in tumor tissues, increasing drug concentration at the target site while minimizing systemic adverse effects [7]. Active targeting involves modifying nanoparticles with specific biomolecules such as ligands, antibodies, or peptides. These molecules selectively bind to overexpressed receptors on cancer cells, promoting better cellular uptake and enhancing the overall therapeutic effect [8].

Gold nanoparticles (AuNPs), due to their unique physicochemical properties, have been used as drug delivery systems (DDSs) and anticancer agents with hyperthermia but also as anti-metastatic agents. Among various AuNPs formulations, glutathione-stabilized AuNPs (Au-GSH NPs) were selected for this study due to their ultrasmall core size (typically below 3 nm), and the use of glutathione—a naturally occurring tripeptide—as a stabilizing ligand contributes significantly to their excellent aqueous dispersibility, low immunogenicity, favorable biodistribution, and ability to inhibit canine osteosarcoma cell extravasation, as previously demonstrated by our team [9]. Importantly, glutathione not only serves as a capping and stabilizing agent but also plays a role in enhancing the redox responsiveness of the nanoparticle surface, potentially enabling environment-sensitive behavior in biological systems. Additionally, the thiol–gold interaction forms strong covalent bonds that ensure high structural stability under physiological conditions.

Recent studies in human cancer models have shown that AuNPs can modulate key processes and aspects of metastasis development, including the epithelial–mesenchymal transition (EMT), matrix metalloproteinases (MMPs), and intracellular signaling pathways such as PI3K/AKT and Hippo. EMT is a crucial process in cancer progression that enables tumor cells to gain migratory and invasive properties [10,11,12]. Li et al. described that AuNPs improved the structure and function of tumor blood vessels, leading to increased blood flow and reduced hypoxia within tumors [13]. Normalizing the tumor vasculature decreased the intravasation of cancer cells and metastases. Additionally, AuNPs directly inhibited EMT by decreasing the expression of mesenchymal markers like vimentin and increasing epithelial markers such as E-cadherin. These combined effects resulted in a significant reduction in lung metastases in human melanoma models treated with AuNPs [13]. Gold nanorods (Au NRs) can inhibit tumor invasion and metastasis by altering the structure of matrix metalloproteinase-9 (MMP-9), a key enzyme in cancer progression. Their effectiveness depends on the aspect ratio, with 3.3 being optimal for the maximum inhibition. Moreover, Au NRs with an aspect ratio of 3.3 suppress X-ray-induced tumor invasion [14]. These findings suggest that Au NRs could be developed as adjustable matrix metalloproteinase (MMP) inhibitors for cancer treatment [14]. In recent studies, Małek et al. demonstrated that Au-GSH NPs effectively inhibit the extravasation of canine OSA cells (from OSCA-8 and OSCA-32 cell lines) using the chick embryo chorioallantoic membrane (CAM) model, indicating their potential as an agent against this crucial step of the metastatic cascade [9].

Furthermore, we previously demonstrated A2M’s role in the metastatic process in canine OSA, and proteomic analysis revealed the significantly lower expression of the A2M protein in two OSA cell lines (OSCA-8 and OSCA-32) compared to osteoblasts. A2M, as a protease inhibitor, is involved in the degradation of extracellular matrix components, a significant process in cancer cell invasion and metastasis [15]. Olbromski et al. investigated A2M expression in human cancers, including breast cancer (IDC), lung cancer (NSCLC), and colorectal cancer (CC), revealing distinct patterns across different malignancies. In IDC, A2M expression was elevated in tumor tissues and correlated with a higher tumor grade and poorer survival, suggesting a role in cancer progression. In NSCLC, A2M expression was decreased in tumor cells compared to non-malignant lung tissue, while its receptor LRP1 was upregulated in the tumor stroma, indicating a potential compensatory mechanism. In CC, A2M was increased in tumor tissues, similarly to IDC, but correlated with better overall survival. These findings highlight the complex role of A2M in cancer biology, with potential implications as a biomarker and therapeutic target, depending on the tumor type [16].

The main aim of this study was to evaluate the influence of Au-GSH NPs on different steps of the metastatic cascade: angiogenesis, migration, and colony formation for canine osteosarcoma. The intermediate aim of this study was to assess whether its mechanism of action is correlated with changes in A2M expression in OSA cells.

## 2. Results

### 2.1. Characterization of Synthesized Au-GSH NPs: Morphology and Colloidal Stability

The morphology and dispersion of the synthesized Au-GSH NPs were evaluated using transmission electron microscopy (TEM). As presented in Figure 1, the nanoparticles exhibit a predominantly spherical shape with a relatively uniform size distribution, reporting an average diameter of 4.3 ± 1.1 nm (Figure 2). A zeta potential of −52.9 ± 10.6 mV was found for Au-GSH NPs, indicating a high surface charge that ensures strong electrostatic repulsion between particles and prevents their aggregation. This value confirms the high colloidal stability of the nanoparticle suspension in aqueous solution [17].

### 2.2. Au-GSH NPs Are Not Cytotoxic to Canine OSA Cells from D-17 Cell Line

D-17 cells treated with 500 µg/mL (the maximum concentration that does not induce Au-GSH NP aggregation) Au-GSH NPs for 24 h under a contrast-phase microscope did not exhibit any signs of changes in morphology in comparison to the non-treated control cells (Figure 3). Both treated and untreated cells exhibit a similar elongated, spindle-like shape with a comparable density and arrangement. There is no visible evidence of structural damage, cell shrinkage, or detachment. Furthermore, the cell mortality assessed using the Countess II automated cell counter with trypan blue exclusion did not exceed 20% at either tested concentration (200 µg/mL and 500 µg/mL) in D-17 cells. For OSCA-32 and OSCA-8 cells, the same evaluations of morphology and viability were applied in a previous study by our team [9]. The analyses showed comparable results: no morphological changes or cell mortality below 20%.

### 2.3. The Influence of Au-GSH NPs on the Migration Rate of the D-17, OSCA-8, and OSCA-32 Cell Lines

An in vitro wound-healing assay was used to analyze the influence of Au-GSH NPs on D-17, OSCA-8, and OSCA-32 cell migration. Au-GSH NPs in both concentrations (200 µg/mL and 500 µg/mL) significantly (*p* ≤ 0.001 after 12 h and *p* ≤ 0.001 after 24 h) inhibited OSA cell migration in comparison to the untreated cells (Figure 4, Figure 5, Figure 6 and Appendix A).

### 2.4. Au-GSH NP Treatment Increases Expression of A2M Protein in Canine OSA Cells

Simple Western analysis revealed a highly significant increase in A2M expression in OSCA-8 cells (*p* ≤ 0.001) and a significant increase in OSCA-32 cells (*p* ≤ 0.05) following treatment with Au-GSH NPs, as compared to the non-treated groups (Figure 7).

### 2.5. Au-GSH NPs Inhibit Angiogenesis in the CAM Assay with Canine OSA Cells

In the chick embryo CAM model, OSCA-32 treated with Au-GSH NPs demonstrated an inhibitory effect on angiogenesis after 72 h (Figure 8F), compared to the negative control with aqua pro injection (Figure 8B) and OSCA-8 alone, where newly formed small vessels and hemorrhagic areas are clearly visible (Figure 8D).

### 2.6. Au-GSH NP Effects on Canine OSA Cell Colony Formation

Au-GSH NPs in a concentration of 200 µg/mL reduced the number of colonies compared to untreated cells in all evaluated canine OSA cell lines (Figure 9B,E,H). Interestingly, the concentration of 500 µg/mL Au-GSH NPs has no or a much lower effect on colony formation in all tested cells (Figure 9C,F,I).

### 2.7. Au-GSH NP Accumulation in OSA Cells

MP-AES was employed to measure nanoparticle uptake by OSA cells. After incubation with Au-GSH NPs, the OSA cells were separated, rinsed, and analyzed to determine their gold content. The average number of nanoparticles absorbed per cell was calculated using the known nanoparticle size and gold content. The rinsing process effectively removed unabsorbed Au-GSH NPs, as confirmed by filtrate analysis. The results, summarized in Table 1, indicate that the OSCA-8, OSCA-32, and D-17 cell lines absorbed Au-GSH NPs.

The data presented in this table shows that the uptake of Au-GSH NPs by OSA cells varies significantly depending on the cell line and the concentration of Au-GSH NPs used. The highest level of absorption was observed in both OSCA-32 cells treated with 200 µg/mL Au-GSH NPs (11.79 ± 2.54%), while the lowest was found in D-17 cells treated with 500 µg/mL Au-GSH NPs (0.47 ± 0.06%) (Table 1).

## 3. Discussion

The inhibition of metastasis is considered the fundamental aim in oncological treatment. AuNPs show promising potential as innovative agents for targeting and disrupting these complex processes. One of the key physical factors that influence the interaction of AuNPs with cancer cells and their efficacy in oncological applications includes their size [18]. Pan et al. determined that the cytotoxicity of AuNPs is size-dependent—small AuNPs (1 to 2 nm in diameter) are highly toxic to cervix carcinoma epithelial cells and melanoma cells, compared to non-toxic larger particles [19]. In this study, we did not observe any toxic effects or morphology changes (Figure 2) of 4.3 ± 1.1 nm at a dose of 500 µg/mL on D-17 OSA cells (cell viability > 80%), which is consistent with our previous research on the OSCA-32 and OSCA-8 cell lines and feline fibrosarcoma cell lines [9,20]. Besides the possible toxic effect, the size of AuNPs significantly impacts their cellular uptake—smaller AuNPs are more readily internalized by cells compared to larger ones [21].

The efficacy of AuNP-based therapies has been evaluated in terms of the steps of the metastatic cascade in various human cancer studies [12,22,23,24]. In this study, we observed the significant (*p* ≤ 0.05 and *p* ≤ 0.001) inhibition of canine OSA cell migration following the treatment with 200 µg/mL and 500 µg/mL AuNPs, respectively, in all tested cell lines (Figure 3, Figure 4 and Figure 5). Our results of the migration assessment are consistent with those described in human oncological research. In human ovarian cancer, 20 nm AuNPs and 35 nm nuclear-targeted AuNPs inhibited the migration and invasion of tumor cells [10,12]. Previously, our team showed that Au-GSH NPs impact MMP2 expression in canine OSA cells [9]. In this study, we demonstrated that Au-GSH NPs increase the expression of A2M in canine OSA cells (Figure 9). A crucial feature of the A2M protein is its capacity to suppress a range of proteases, including MMPs involved in extracellular matrix remodeling [25,26]. Our study is the first one that demonstrates the influence of Au-GSH NPs on the increase in A2M expression in OSA cells, indicating that A2M’s modulating mechanism may be one of multiple interacting mechanisms of Au-GSH NPs’ anti-metastatic properties. Further studies, including those on the role of Au-GSH NPs in *A2M* and *ADAMTS1* gene expression and their potential application in anti-metastatic therapies, are needed in vitro and in vivo.

Another integral part of the metastatic cascade is angiogenesis. This process enables the development of new blood vessels in the primary tumor. This mechanism is activated in response to hypoxia in the tumor microenvironment, leading to the production of vascular growth factors such as VEGF, which stimulate the formation of pathologically permeable vessels, promoting both tumor growth and the ability of tumors to metastasize [27]. The chorioallantoic membrane (CAM) model of chick embryos has been widely used for many years in tumor angiogenesis research [28]. Our study demonstrated the antiangiogenic properties of 500 µg/mL Au-GSH NPs in a canine OSA model (Figure 8). The inhibition of angiogenesis may be linked to molecular mechanisms previously described for AuNPs, including the suppression of vascular endothelial growth factor (VEGF) expression, the inhibition of VEGF receptor (VEGFR) signaling pathways, a reduction in MMP activity, and the downregulation of key angiogenesis-related pathways such as PI3K/Akt and MAPK/ERK. These mechanisms impair endothelial cell proliferation, migration, and new vessel formation, contributing to the antiangiogenic effect [29].

Besides the metastatic cascade steps, the phenomenon of tumor proliferation (uncontrolled cell division) is a characteristic of cancer and an important driver of its progression. Proliferation directly impacts tumor growth, progression, and the development of metastases, making it an important focus in cancer research [30]. The colony formation assay shows the ability of cancer cells to undergo clonal expansion and form a colony, providing insights into the proliferative capacity of cancer cells and the efficacy of treatments [31]. Pendiuk et al. evaluated the negative correlation between the clonogenic capacity of human melanoma cells and the treatment with ultrasmall AuNPs (~3 nm) stabilized by the anionic polysaccharide gum arabic (GA-AuNPs) [32]. In our study, the observed effect of Au-GSH NPs (Figure 9) contrasts with the dose-dependent correlation reported by Pendiuk and collaborators [32]. Our findings reveal a more pronounced inhibition of colony formation in canine OSA cells at a dose of 200 μg/mL compared to 500 μg/mL (Table 1), suggesting a non-linear dose–response relationship. The results may be directly related to the higher efficiency of Au-GSH NP uptake by OSA cells at the dose of 200 μg/mL than at the dose of 500 μg/mL (Table 1). The highest Au-GSH NP accumulation visible for the OSCA-32 cell line at a concentration of 200 μg/mL corresponds to the highest inhibition of colony formation (Figure 9H). However, the direct mechanism underlying the more effective inhibition of colony formation at the lower dose of Au-GSH NPs in canine OSA cells remains unknown and needs further exploration. Furthermore, the formation of a protein corona on the surface of Au-GSH NPs cannot be excluded. When nanoparticles enter biological fluids, they rapidly adsorb various proteins, forming a dynamic protein corona that can significantly influence their biological identity and interactions with cells. The composition and structure of this corona are known to affect cellular uptake mechanisms, biodistribution, and overall therapeutic efficacy. However, the specific role of the protein corona in the uptake of AuNPs by cancer cells remains unclear and is not fully elucidated in the current literature. Therefore, further studies are warranted to investigate the composition of the protein corona formed on AuNPs under different physiological conditions and to understand how it influences their interaction with cancer cells.

To the best of our knowledge, the present study is the first to demonstrate the influence of Au-GSH NPs on canine OSA cells’ migration, angiogenesis, and colony formation. However, the limitations of this study include the small number of cell lines analyzed during testing. The cell lines used in this study represent osteosarcoma models with varying degrees of aggressiveness and are the only commercially available canine OSA cell lines. Based on the available literature and molecular profiling, OSCA-8 is considered to exhibit a highly aggressive phenotype, while OSCA-32 and D-17 are classified as moderately aggressive [33,34]. Additional derived OSA cell lines with confirmed in vivo metastatic capacity, such as Abrams or TOT, should be included in future investigations to enhance the reliability of the obtained results and increase the power of statistical analyses [35]. Additionally, this study was restricted to in vitro and in ovo models, without in vivo validation, and focused only on selected steps of the metastatic cascade. It is necessary to emphasize that these are preliminary results, and further investigations are required to fully understand the therapeutic potential of Au-GSH NPs and their mechanism of anti-metastatic action in canine osteosarcoma.

## 4. Materials and Methods

### 4.1. Cell Culture

Three canine OSA cell lines, OSCA-8, OSCA-32 (Kerafast, Boston, MA, USA), and D-17 (ATCC, Manassas, VA, USA), and a normal canine osteoblast cell line—CnOb (Cell Applications, San Diego, CA, USA)—were used. The OSCA-8 cell line was derived from a left shoulder tumor of an intact 1-year-old male Rottweiler. The OSCA-32 cell line was derived from a tumor in the left wrist of a 9-year-old spayed female Great Pyrenees. The D-17 cell line was derived from an osteosarcoma metastatic to the lung of an 11-year-old female poodle.

Osteosarcoma cells (OSCA-8, OSCA-32, and D-17) as well as CnOb were cultured under aseptic conditions in a sterile chamber with laminar airflow, model ESCO Airstream AC2-3E8 (ESCO, Warsaw, Poland). They were maintained in Dulbecco’s Modified Eagle Medium (DMEM, Gibco, Waltham, MA, USA), supplemented with 10% fetal bovine serum (FBS, Life Technologies, Gibco, Waltham, MA, USA), antibiotics (Primocin—InvivoGen, Waltham, MA, USA—and penicillin–streptomycin 100× solution—HyCloneTM, Marlborough, MA, USA), and 1% HEPES buffer (HyCloneTM, Marlborough, MA, USA). The cultures were incubated at 37 °C with 5% CO_2_ and 95% humidity. Experiments were conducted when the cells reached 70–80% confluence during the logarithmic growth phase. Cell viability was measured using trypan blue with an Invitrogen Countess II automated cell counter (Thermo Fisher, Waltham, MA, USA).

### 4.2. Au-GSH NP Synthesis

The beginning of synthesis was performed at room temperature (RT) by combining 4 mL of hydrogen tetrachloroaurate (III) trihydrate solution (HAuCl_4_ × 3 H_2_O) (Sigma-Aldrich, St. Louis, MO, USA) at a concentration of 126.97 mmol/dm^3^ with 40 mL of distilled water in a 250 mL flask. The mixture became yellow in color. Next, by gently mixing with a magnetic stirrer placed at the bottom of the flask, 270 mg (0.88 mmol) of GSH (Sigma-Aldrich, St. Louis, MO, USA) in its reduced form was added. The mixture became orange-brown in color, and after a few minutes it became colorless. This was due to the fact that gold had undergone an initial reduction from the +III to +I oxidation state. Mixing was continued at RT, and after about an hour, the appearance of white turbidity was observed at the surface. This was caused by a decrease in pH, resulting in the precipitation of the unreacted GSH residue by the protonation of its carboxyl groups. To remove the turbidity, 5 mL of alkaline sodium bicarbonate (NaHCO_3_) (Thermo Fisher, Waltham, MA, USA) was added to the mixture. This caused the reionization of GSH carboxyl groups and its redissolution in water. When the mixture became clear again, 83 mg (2.19 mmol) of sodium borohydride (NaBH_4_) (Sigma-Aldrich, St. Louis, MO, USA) solution freshly dissolved in 11 mL of cold distilled water was added as a reducing agent. As a result, the mixture turned into a dark maroon color. This was due to the reduction of gold atoms from the +I to 0 oxidation state. After 2 h of mixing the mixture at RT, 26 mL of methanol (MeOH) (Sigma-Aldrich, St. Louis, MO, USA) was added to precipitate 4 nm structures. The mixture was placed into Falcon-like test tubes and centrifuged for 9 min, 8000 RPM, RT. The precipitate, containing larger structures, was discarded. Another portion of 53 mL of MeOH was added to the supernatant and centrifuged again, in the same conditions. This was conducted to precipitate structures of 2 nm in size. The supernatant, containing unreacted substrates and smaller structures, was discarded. The precipitate was suspended in distilled water. The suspension was placed in a dialysis membrane with 3500 MWCO pores (Spectra/Por 3, 45 × 29 mm). The membrane was placed in a crystallizer with spout, 500 mL, 115 mm (DURAN, Wertheim, Germany) filled with distilled water. The suspension was mixed using a magnetic stirrer for 3 days to purify it from residual impurities. Water was exchanged every day. After dialysis was complete, the membrane content was placed in the Falcon-like test tube, and an equal volume of PBS (Na_2_HPO_4_-KH_2_PO_4_) buffer (Sigma-Aldrich, St. Louis, MO, USA) was added, obtaining a clean Au-GSH structure. In order to measure the molecular concentration, three test tubes were prepared. Every tube was weighed 3 times on a precise scale. A total of 1 mL of the Au-GSH structure suspension was placed in each tube, using an automatic pipette. Tubes were then placed in an oven and dried for approx. 2 h at over 100 °C. After the water evaporated completely, the tubes were weighed again. The difference before and after the drying process was stated as the weight of the Au-GSH structure. TEM images (EM-1400Flash, JEOL Co., Tokyo, Japan) confirmed the proper size, shape, and dispersion of nanoparticles, and zeta potential (Zetasizer Nano-ZS, Malvern Panalytical, Malvern, Worcestershire, UK) showed the physicochemical properties of the nanoparticles’ surface and the stability of the colloid.

### 4.3. Migration Rate Assessment in Canine OSA Cell Lines Treated with Au-GSH NPs

Cells were seeded into culture inserts (2-well Ibidi culture inserts, Ibidi GmbH, Grafelfing, Germany) at a density of 3 × 10^4^ cells per well. Once the cells reached approximately 90% confluence, the medium was replaced with one containing Mitomycin C (Abcam, Cambridge, UK)—a DNA synthesis inhibitor used to assess cell migration—at a final concentration of 10 μg/mL, ensuring the minimal loss of viability while effectively inhibiting cell division. The pre-treatment with Mitomycin C stops cell proliferation, allowing for a true evaluation of cancer cell migration by excluding the impact of proliferating cells on wound closure [36]. The cell monolayers were incubated at 37 °C under standard conditions. After 3 h, the Mitomycin C medium was replaced with a serum-free medium, enriched with AuNPs at concentrations of 0 µg/mL (control group), 200 µg/mL, and 500 µg/mL, and the culture inserts were removed. For adherent cells, like those in this study, it is essential that the cells attach to the bottom of the wells and that proliferation is halted to assess migration properly. Cell migration within the scratch was observed at specific time intervals (t0 = 0 h; t1 = 12 h; t2 = 24 h) using an inverted microscope (Primovert, Zeiss, Munich, Germany) at 4× magnification. The images captured were later analyzed using Zen Pro 2012 (Zeiss, Munich, Germany), and cancer cell migration was quantified by measuring the distance between the scratch edges (100 measurements per scratch) following the method described by Rodriguez et al. [37]. This experiment was conducted in triplicate.

### 4.4. Au-GSH NP Effects on A2M Expression in OSA Cells

#### 4.4.1. Protein Isolation, Cleaning, and Precipitation

Proteins from cell pellets (OSCA-8, OSCA-32 treated and untreated with 500 µg/mL Au-GSH NPs (*n* = 7 per group), and CnOb (positive control)) were homogenized and isolated using a solution composed of RIPA buffer (Sigma-Aldrich, Saint Louis, MO, USA) combined with a protease inhibitor cocktail (Sigma-Aldrich, Saint Louis, MO, USA), in a 1:100 ratio. Specifically, the cell suspension was centrifuged at 300× *g* for 3 min at room temperature (RT) and resuspended in 1.5 mL of phosphate-buffered saline (PBS) (Gibco, Waltham, MA, USA). The suspension obtained from a single culture flask was split into three 0.5 mL samples and centrifuged again (300× *g*, 3 min, RT). After removing the supernatant, 200 µL of this prepared solution was added to the pellet. The mixture was vortexed three times for 5 s at 5 min intervals and subsequently centrifuged at 16,400× *g* for 15 min at 4 °C. Protein concentrations in the homogenates were assessed using the bicinchoninic acid (BCA) assay. A BCA reagent was prepared following the instructions from the BCA Protein Assay Kit (Abcam, Cambridge, UK). For the analysis, 200 µL of the BCA reagent and 10 µL of protein homogenate were added to each well of a 96-well plate, with each sample measured in triplicate. A standard curve was prepared using a bovine serum albumin (BSA, Sigma-Aldrich, Saint Louis, MO, USA) stock solution at an initial concentration of 1 mg/mL. The standard curve included the following BSA/RIPA solution concentrations: 0, 200×, 400×, 600×, 800×, and 1000×. Spectrophotometric readings were performed at 37 °C with a wavelength of 562 nm using the Infinite 200 PRO M Nano Plate Reader (Tecan, Zurich, Switzerland), operated with Tecan I-Control 2.0 Software (Tecan, Zurich, Switzerland).

#### 4.4.2. Quantitative Protein Expression Analyses

The expression of A2M was assessed by the use of the Simple Western technique with total protein normalization in a Jess analyzer (Protein Simple, Minneapolis, MN, USA). Reagents from the EZ Standard Pack (Bio-Techne, Minneapolis, MN, USA) were prepared following the Jess protocol. The EZ Standard Pack 12–230 kDa includes a pre-prepared biotinylated ladder, a 5× fluorescent master mix, and a DTT solution. To prepare a 400 mM DTT solution, 40 µL of deionized water was combined with DTT in a clear tube. The master mix was then combined with 20 µL of 10× sample buffer and 20 µL of the prepared DTT solution. Protein homogenates were diluted with 0.1× sample buffer (Bio-Techne, Minneapolis, MN, USA) to achieve the appropriate protein concentration for analysis, which varies based on the expression level of the target protein. A 5× fluorescent master mix was diluted with the lysate at a 1:4 ratio in a microcentrifuge tube. Samples were denatured at 95 °C for 5 min, then vortexed, spun at 3000 rpm for 10 s, and placed on ice.

The primary anti-human A2M antibody (MAB1938, Bio-Techne, Minneapolis, MN, USA) was diluted using Milk-free Antibody Diluent (Bio-Techne, Minneapolis, MN, USA). A luminol-S and peroxide mixture (1:1 ratio) was vortexed and kept on ice. All reagents, including the secondary anti-mouse antibody (Bio-Techne, Minneapolis, MN, USA), were applied to the 12–230 kDa separation module in accordance with the manufacturer’s instructions. The primary antibody saturation was 1:6.25 and at a concentration of 0.2 mg/mL according to the previously published protocol for A2M expression in canine OSA cells [33]. Total protein expression levels were normalized using the total protein detection module, ensuring the accurate quantification of protein levels across samples. The obtained data were analyzed using the Compass for Simple Western software (version 7.0, Bio-Techne, Minneapolis, MN, USA). The results were analyzed using the following criteria: total protein area under the curve differences between samples < 20% (Appendix A), signal-to-noise ratio > 10, and signal-to-background ratio > 3.

### 4.5. Au-GSH NP Effects on Angiogenesis—Chick Embryo Chorioallantoic Membrane (CAM) Assay

Fertilized eggs of the domestic chicken (*Gallus gallus*) (Ross 308) (*n* = 30) were incubated under the following conditions: 37 °C and 70% humidity in a Fest Midi I incubator (F.U.H. Walenski, Gostyn, Poland) according to the previously described protocol with modifications [38]. Briefly, on the sixth day of incubation, a small window was made in each eggshell, and a sterile silicon ring (7 mm in external, 5 mm in internal diameter, 2 mm thick) was placed onto the CAM under aseptic conditions. The chick embryos were divided into three groups (*n* = 10 per group). For each group, the medium with 5 × 10^6^ untreated OSA cells, OSA cells treated with Au-GSH NPs (prepared as described in Section “Preparation of OSA Cells Treated with Au-GSH NPs”) for 24 h, and an aqua pro injection (control group) was injected into the silicone rings. Embryo survival was assessed by using an ovoscope after 24 and 48 h, with an average viability of 50%. Angiogenesis was observed at specific time intervals (t0 = 0 h; t1 = 72 h) using a VHX-900F lightning digital microscope (Keyence International Bedrijvenlaan, Mechelen, Belgium) at 4× magnification. Images were captured with VHX-900F Ver 1.6.1.0 software (Keyence International Bedrijvenlaan, Mechelen, Belgium).

#### Preparation of OSA Cells Treated with Au-GSH NPs

OSCA-32 cells were plated in 6-well plates (Becton Dickinson, Franklin Lakes, NJ, USA) at a density of 5 × 10^5^ cells per well. Once the cells reached approximately 70% confluence, they were treated with Au-GSH NPs (500 µg/mL, 1 mL per well) for 24 h. Following the treatment, the cells were collected, washed twice with PBS, and prepared for subsequent experiments. The 500 µg/mL Au-GSH NP concentration used in this study was selected based on previously published results with Au-GSH NP extravasation efficacy for OSA cells used on the CAM model [9].

### 4.6. Au-GSH NP Effects on Canine OSA Cell Colony Formation

OSCA-8, OSCA-32, and D-17 cells were seeded at a density of 5 × 10^3^ cells per well into 6-well plates. A total of 24 h later, the medium was replaced with AuNP-containing medium at concentrations of 200 µg/mL and 500 µg/mL and one without AuNPs (negative control), and cells were incubated for 24 h in standard conditions. After that, the medium was replaced with standard culture medium, and cells were maintained until the seventh day of the experiment. Then the cells were fixed with methanol in two steps: a 12 min incubation followed by a 70 min incubation at room temperature, rinsing using ultra-pure water after each step. Crystal violet staining (0.1% in 95% ethanol) was applied and left overnight. Excess stain was rinsed off with ultra-pure water, and staining was repeated for enhanced visibility. All experiments were repeated in triplicate. Photographs of the stained wells were captured using a D3500 camera (Nikon, Tokyo, Japan) to analyze the nanoparticle effects on cell growth [31].

### 4.7. Accumulation of Au-GSH NPs in Canine OSA Cells by Microwave Plasma Atomic Emission Spectroscopy (MP-AES)

OSCA-8, OSCA-32, and D-17 cells were seeded into 25 cm^2^ flasks for adherent cell culture (Thermo Fisher, Waltham, MA, USA). A total of 24 h later, the medium was replaced with Au-GSH NP-containing medium (5 mL medium per flask, at concentrations 200 and 500 µg/mL), and cells were incubated for 24 h. To determine the number of AuNPs absorbed by the cells, first, the cells were mineralized using concentrated nitric acid (65%, Avantor, Gliwice, Poland). Then, concentrated hydrochloric acid (36%, Avantor, Gliwice, Poland) was added. Mixing nitric acid with hydrochloric acid in a 1:3 volume ratio produced aqua regia, resulting in a transparent yellow solution. This solution was then analyzed using MP-AES (MP-AES Agilent 4200, Tokyo, Japan). An analytical standard solution (SCP Science, Baie-D’Urfe, QC, Canada; LOT: S160615016, 1001 ± 4 µg/mL) was used as a reference material and appropriately diluted to create a 5-point calibration curve. Each calibration point and measurement were repeated three times. The determined concentration was then used to calculate the number of nanoparticles per cell, assuming that the density of Au-GSH NPs is equivalent to that of bulk gold (19.28 g/cm^3^), the average nanoparticle radius is 2.15 nm, and the number of cells in the mineralized sample is known.

### 4.8. Statistical Analyses

Statistical analyses were performed using GraphPad Prism version 9.0 (GraphPad Software, La Jolla, CA, USA) with data presented as the mean ± standard deviation (SD) and statistical significance set at *p* ≤ 0.05. For the A2M protein expression levels of untreated and AuNP-treated osteosarcoma cells, a two-tailed unpaired Student’s *t*-test was used. A one-way analysis of variance (ANOVA) followed by Tukey’s multiple comparison post hoc test was applied to assess statistical differences between each group for cell migration. Statistical significance was marked with an asterisk at *p* ≤ 0.05 (*), and high statistical significance was marked at *p* ≤ 0.01 (**) and very high statistical significance was stated as *p* ≤ 0.001 (***).

### 4.9. Use of Generative AI

Generative artificial intelligence (GenAI) tool (Chat GPT 4.o, since March 2024) was used to assist in the preparation of this manuscript by combining individual images into composite figures (Figure 3, Figure 8 and Figure 9). Grammarly Pro software was used to check the style and grammar of the English used.

The final version of this manuscript was manually checked and approved by the authors.

## 5. Conclusions

Au-GSH NPs exhibit inhibitory effects on key tumorigenic processes in canine OSA cells (OSCA-8, OSCA-32, and D-17): migration, angiogenesis, and proliferation. Additionally, the inhibition of colony formation was more pronounced at 200 µg/mL than at 500 µg/mL, which may be related to the higher Au-GSH NP cell internalization at the lower concentration. This is the first study demonstrating the influence of Au-GSH NPs on the increase in A2M expression, which may be related to the anti-metastatic efficacy of Au-GSH NPs; however, further analyses are required to elucidate the exact mechanisms underlying these observations. Furthermore, Au-GSH NPs effectively suppressed angiogenesis in the CAM model, further supporting their potential to limit metastatic progression. These preliminary findings highlight the therapeutic potential of Au-GSH NPs in canine osteosarcoma treatment; however, further in vivo studies are needed to assess their clinical relevance in veterinary oncology.

## Figures and Tables

**Figure 1 ijms-26-06102-f001:**
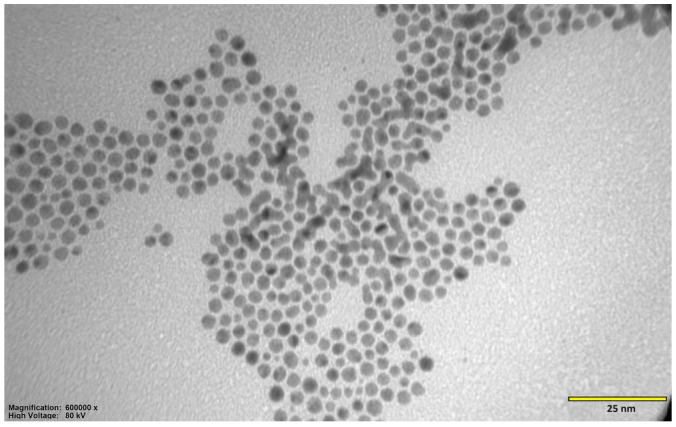
TEM images of Au-NPs stabilized with glutathione (Au-GSH NPs). The nanoparticles exhibit a spherical shape and uniform size distribution. No significant aggregation was observed, indicating the effective stabilization of the colloidal system. Scale bars: 25 nm.

**Figure 2 ijms-26-06102-f002:**
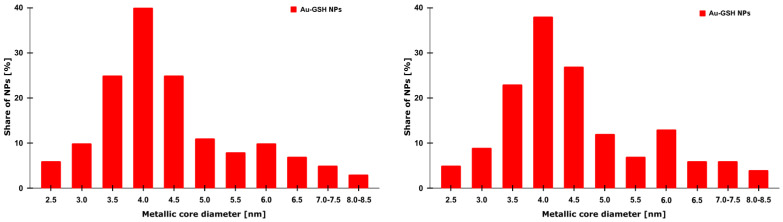
Size distribution histograms of the metallic core of Au-GSH nanoparticles (Au-GSH NPs). The diameters were estimated based on the scale bars visible in the respective TEM images using ImageJ software (version 1.51, 23 April 2018). The quantitative analysis illustrates the distribution of nanoparticle sizes within the examined sample.

**Figure 3 ijms-26-06102-f003:**
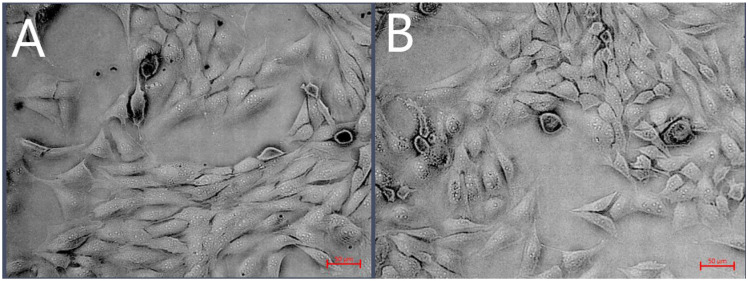
Phase-contrast microscopy images of D-17 cells treated with 500 µg/mL Au-GSH NPs (**A**) vs. non-treated (**B**) D-17 cells, shown at 20× magnification, scale bar 50 µm.

**Figure 4 ijms-26-06102-f004:**
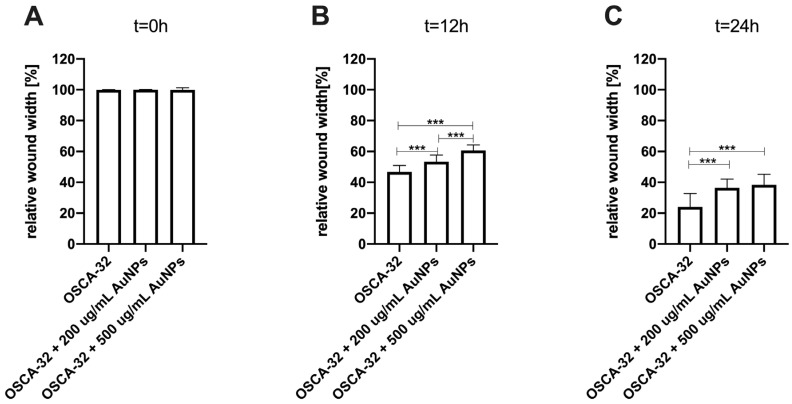
Bar graphs showing Au-GSH NP effects in two concentrations (200 µg/mL and 500 µg/mL) on inhibiting the migration of canine osteosarcoma cells from the OSCA-32 cell line at (**A**) t0 (0 h), (**B**) t1 (12 h), and (**C**) t2 (24 h). *** *p* ≤ 0.001.

**Figure 5 ijms-26-06102-f005:**
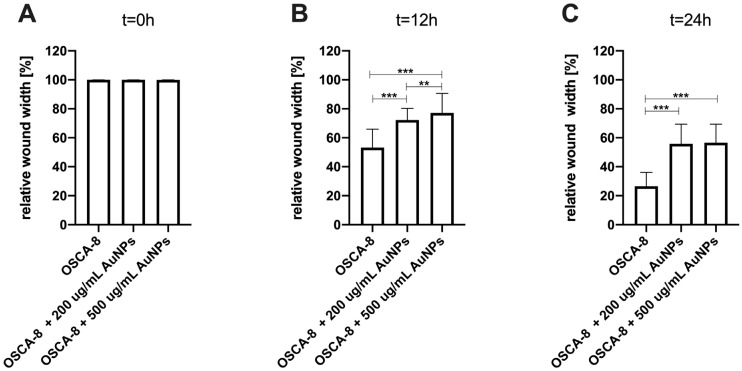
Bar graphs showing the Au-GSH NP effects in two concentrations (200 µg/mL and 500 µg/mL) on inhibiting the migration of canine osteosarcoma cells from the OSCA-8 cell line at (**A**) t0 (0 h), (**B**) t1 (12 h), and (**C**) t2 (24 h). *** *p* ≤ 0.001, ** *p* ≤ 0.01.

**Figure 6 ijms-26-06102-f006:**
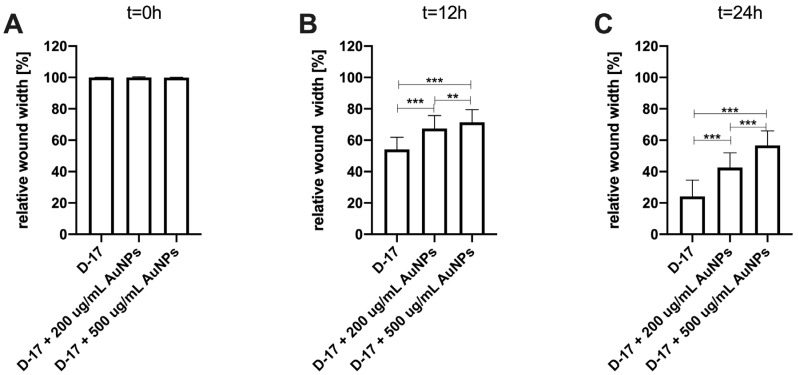
Bar graphs showing the Au-GSH NP effects in two concentrations (200 µg/mL and 500 µg/mL) on inhibiting the migration of canine osteosarcoma cells from the D-17 cell line at (**A**) t0 (0 h), (**B**) t1 (12 h), and (**C**) t2 (24 h). *** *p* ≤ 0.001, ** *p* ≤ 0.01.

**Figure 7 ijms-26-06102-f007:**
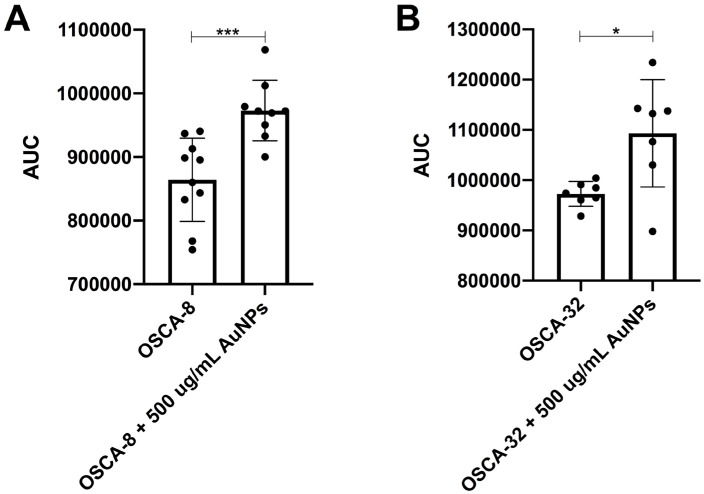
Bar graphs showing A2M expression in OSCA-8 (**A**) and OSCA-32 (**B**) cells following treatment with Au-GSH NPs. A highly significant increase in A2M expression was observed in OSCA-8 cells (*** *p* ≤ 0.001), and a significant increase was detected in OSCA-32 cells (* *p* ≤ 0.05), compared to untreated control groups.

**Figure 8 ijms-26-06102-f008:**
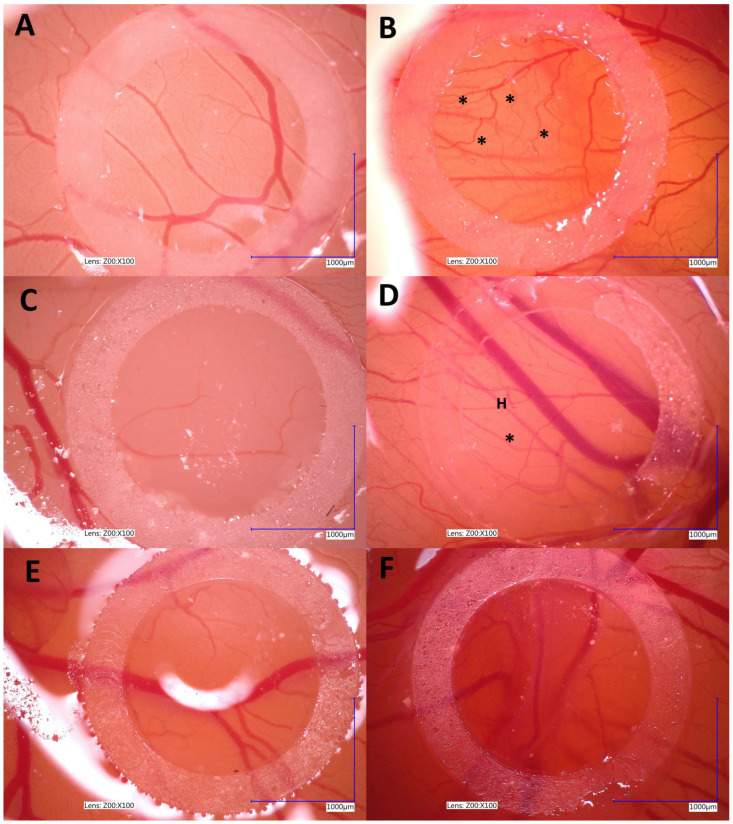
Representative images of angiogenesis in the CAM assay following treatment with OSCA-32 incubated with 500 µg/mL Au-GSH NPs. CAMs from the control group at T = 0 h (**A**) and T = 72 h (**B**). CAMs treated with OSCA-32 cells at T = 0 h (**C**) and T = 72 h (**D**). CAMs treated with OSCA-32 incubated with 500 µg/mL AuNPs at T = 0 h (**E**) and T = 72 h (**F**). Newly formed blood vessels are indicated by an asterisk (*), and hemorrhagic areas are marked with the letter H.

**Figure 9 ijms-26-06102-f009:**
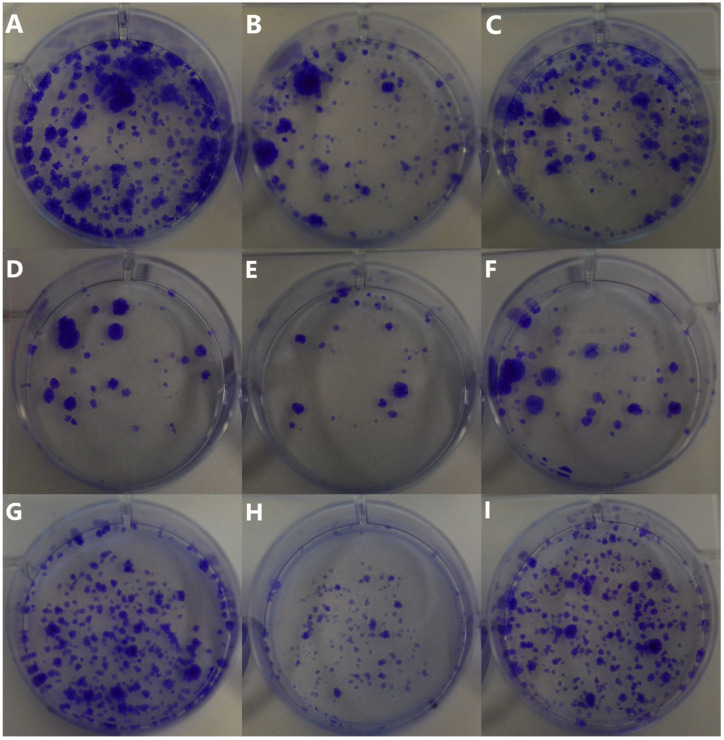
Effects of Au-GSH NPs on OSA cell colony forming after t = 72 h. Untreated D-17 cells (**A**), D-17 cells treated with 200 µg/mL Au-GSH NPs (**B**) and 500 µg/mL Au-GSH NPs (**C**); untreated OSCA-8 cells (**D**), OSCA-8 cells treated with 200 µg/mL Au-GSH NPs (**E**) and 500 µg/mL Au-GSH NPs (**F**); untreated OSCA-32 cells (**G**), OSCA-32 cells treated with 200 µg/mL Au-GSH NPs (**H**) and 500 µg/mL Au-GSH NPs (**I**).

**Table 1 ijms-26-06102-t001:** Uptake of Au-GSH NPs by OSA cells.

Cell Line	Number of Au-GSH NPs/Cell	Au-GSH NPs Absorbed, %
OSCA-8 + Au-GSH NPs 200 µg/mL	1.80 × 10^6^ ± 2.15 × 10^6^	10.51 ± 5.99
OSCA-8 + Au-GSH NPs 500 µg/mL	3.05 × 10^5^ ± 2.53 × 10^5^	4.42 ± 1.85
OSCA-32 + Au-GSH NPs 200 µg/mL	6.32 × 10^6^ ± 5.74 × 10^5^	11.79 ± 2.54
OSCA-32 + Au-GSH NPs 500 µg/mL	4.47 × 10^6^ ± 7.74 × 10^5^	5.22 ± 0.91
D-17 + Au-GSH NPs 200 µg/mL	2.51 × 10^6^ ± 2.32 × 10^6^	2.38 ± 0.82
D-17 + Au-GSH NPs 500 µg/mL	2.47 × 10^6^ ± 8.98 × 10^5^	0.47 ± 0.06

## Data Availability

On request, the raw data obtained in this study can be obtained from the authors (S.S.W., M.W. (Marek Wojnicki), M.W. (Michał Wójcik) and K.A.Z.-K.).

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
