# Peer review of "The Anti-Metastatic Properties of Glutathione-Stabilized Gold Nanoparticles—A Preliminary Study on Canine Osteosarcoma Cell Lines"

_ijms, 2025, doi:10.3390/ijms26136102_

Round 1
Reviewer 1 Report
Comments and Suggestions for Authors
In the submitted manuscript titled “Anti-metastatic properties of glutathione-stabilized gold nanoparticles – a preliminary study on canine osteosarcoma cell lines”, the authors revealed part of the potential ability of gold nanoparticles (AuNPs) as anti-metastatic agents in canine osteosarcoma (OSA) by using several cell lines. The authors' findings, while partly observational, are generally supported by appropriate data and, overall, seem to be consistent with the aims and scope of the journal, IJMS.
>Contrary to the title, the term “glutathione-stabilized gold nanoparticles” is used too infrequently and AuNPs is used instead. Please unify the overall wording.
Abstract
>Lines 38-41
It is certainly an interesting phenomenon, but the reasons for them are even more interesting. Please provide the author's ideas.
Introduction
>Lines 92-94
Please add references.
>Lines 96-105
Please specify the animal species of breast cancer (IDC), lung cancer (NSCLC), and colorectal cancer (CC).
>Please clarify the merits of using Au-GSH in the Introduction section. Why did the authors select Au-GSH among AuNPs tools in this study?
Results
>Line 116
Please show quantitative data on size distribution.
>Line 127
Here, D17 cell line is mentioned for the first time in this manuscript. Please specify that this cell line is derived from canine OSA.
>Line 129, 183, 194
Please clarify the definition of control. Non-treated of AuNP? normal canine osteoblast cell line?
>Lines 132-133
Please add data on cell viability. Additionally, what about the other cell lines?
>Lines 181-183
This study presents A2M amounts corrected for total protein, but the reliability in this comparison analysis is questionable. Please provide data in which the use of AuNPs does not result in a change in total protein per cell or correction by an inner control.
>Lines 193-196
Is quantification possible for this examination?
Discussion
>Lines 251-286
The authors provide a lengthy description of factors other than A2M, which is the only factor analyzed in this study. Especially in Lines 254-278, the topic of human and canine tumors is mixed and may lead the reader to mistakenly assume that all factors mentioned in the manuscript are involved. Please make significant revisions to the manuscript to simply discuss the considerations that can be drawn from the results of this study, and also move the description of the other factor’s relevance into the other parts (for example, Introduction or limitation).
>332-334
Please provide details as to why the cells used in this study are not sufficient. What cell lines should be included in further studies? Please discuss this issue in conjunction with the properties of the cell lines used in this study.
Figure 1
The images presented by the authors are inadequate to estimate the shape and size of the AuNPs. Please replace them with more highly enlarged images.
Figure 2
Visual evidence on cellular uptake would help readers directly understand the dynamics of AuNPs (in fact, they may be superficial bonding). Do the authors perform observations with polarizing microscope?
Figures 3-5
Wound healing assay data would be more visually and objectively indicative of the inhibitory effect if presented in terms of healing rates (%) rather than actual measured values. Furthermore, line graphs, as in the paper (https://doi.org/10.1016/bs.mie.2020.04.011), are more appropriate for the present study. Please revise these, including statistical analysis.
Figures 3-8
For simplicity in presenting information, relevant figures should be merged (for example, Figures 3 and 6).
Reviewer 2 Report
Comments and Suggestions for Authors
Comments:
The article “Anti-metastatic properties of glutathione-stabilized gold nanoparticles – a preliminary study on canine osteosarcoma cell lines” by Wilk et al. evaluated the anti-metastatic properties of AuNPs through their influence on OSA cell migration, proliferation, and colony formation in vitro, as well as their anti-angiogenic properties on the chick embryo chorioallantoic (CAM) model. The author has demonstrated that AuNPs significantly inhibited migration and colony formation in canine osteosarcoma cells. Additionally, they also reported the significant increase of A2M expression in cancer cells after AuNPs treatment. The data presented is comprehensive and indicated the therapeutic potential of AuNPs in canine osteosarcoma treatment.
Other comments:
- Figure 5 C, typo “t = 12h” should be “t = 24h”
- How is the stability of this AuNPs suspension over time? Does it have to be freshly prepared?
Round 2
Reviewer 1 Report
Comments and Suggestions for Authors
I appreciate the authors' response to my previous comments. I have no further comments for this manuscript.